

# Comment on: "Spatial characterization of long-term hydrological change in the Arkavathy watershed adjacent to Bangalore, India" by Penny et al. (2018)

Nitin Bassi[1], B.K. Harish Kumara[2], Meera Sahasranaman[3], Arijit Ganguly[3]

[1]Institute for Resource Analysis and Policy, Liaison Office, Delhi-110085, India
[2]Center for Education Environment and Community, Hassan-573130, Karnataka, India
[3]Institute for Resource Analysis and Policy, Head Office, Hyderabad-500082, India

*Correspondence to*: Nitin Bassi (nitinbassi@irapindia.org)

**Abstract.** This article is a critique on the paper 'Spatial characterization of long-term hydrological change in the Arkavathy watershed adjacent to Bangalore, India', by Penny et al. (2018), published in the journal of Hydrology and Earth System Sciences (volume 22, issue number 1). Penny et al. (2018) have applied a methodology for analysing long-term hydrological changes by treating a 'gauged watershed' as a 'un-gauged' one. The arguments presented in this article highlights choice of inappropriate methodology (for a watershed which is actually gauged) and faulty assumptions by Penny et al. (2018) for analysing the watershed scale hydrological changes, generating misleading results and inferences.

## 1 Introduction

Globally there is an increase in demand for water, more so in countries located in semi-arid to arid regions. In India, which is largely semi-arid, many large river basins such as Krishna, Cauvery and Pennar are on the verge of closure or are already closed, with no discharge into the natural sinks (oceans) in low rainfall years. In order to manage water in such basins, it is important to first identify the factors causing growth in demand for water and also the long-term changes in water supply from catchments and aquifers (runoff and groundwater). The long-term changes in runoff can mainly be because of rainfall change, land use changes (leading to increased evapo-transpirative demand causing reduction in runoff generation), and groundwater intensive use in areas where aquifer discharge contributes to runoff in the form of base flows. Changes in groundwater availability from natural recharge can be caused by changes in infiltration of water from precipitation and natural discharge from aquifers. Apart from understanding the aggregate level changes, it is also important to analyse the spatial patterns, especially in the upstream and downstream parts of the basin as most of the time it is the people in the latter that get affected by the hydrological changes occurring upstream.

One such attempt to explain spatial pattern of long-term hydrological changes has been made by Penny et al. (2018) in a watershed named Arkavathy in Cauvery river basin. The watershed lies in the state of Karnataka. The study asserts to have used a methodology which is appropriate for ungauged catchments. However, in reality Arkavathy is a gauged watershed.





Thus, the study methodology, assumptions, results and inferences have serious shortcomings, findings based on which have limitations when it comes to making useful policy inferences. The major contentious issues are discussed in the subsequent sections of the article.

## 2 Concerns about data unavailability

A few recent studies that have focused on water management issues in India, have reported lack of time series data sets on hydrology and geo-hydrology and questioned the reliability of data available from official agencies (see for instance, Srinivasan et al., 2015; Penny et al., 2018). Often, existing resources with the official agencies to process and manage data properly are debated among the researchers. In some cases, raw data collected directly from the farmers based on recall method on the phenomenon of groundwater depletion that started occurring as long back as 40-45 years is considered to be

more reliable by researchers. While validated data might not be available for free on websites, they are certainly available with the central and state agencies in charge of monitoring surface water flows and groundwater in the basin. Researchers only need to reach out to the concerned agencies and, collect and interpret the data in the best possible way.

Moreover, the situation with regard to hydrological data availability and their reliability has improved remarkably over the past 10-15 years in India. After the World Bank supported Hydrology Project (initiated in 1995), most of the historical data

related to meteorology, hydrology, reservoir operations, water quality and groundwater are digitised in many states through establishment of Water Data Centres (WDC). Presently, nine major states in India, including Karnataka where the Arkavathy watershed studied by Penny et al. (2018) falls, have fully operational WDC.

The authors have raised a concern that official data on surface water flows and groundwater in the watershed were not available and justified the use of Remote Sensing (RS)/GIS based methodology which they claim is suitable for ungauged

catchments (page 596 and 605 of Penny et al., 2018). It is important to mention here that in India hydrological, meteorological and groundwater data are collected and maintained by both National (central) and Provincial (state) agencies. The gauging or observations stations managed by central agencies are different from those maintained by state agencies. As per the basin report by the Ministry of Water Resources, River Development and Ganga Rejuvenation (Government of India, 2014), middle part of Cauvery river basin having an area of about 57,281 sq. km (where Arkavathy watershed falls) has 20

hydrological observations sites, 159 meteorological stations and 612 groundwater observation wells which are maintained by central agencies alone (Fig. 1). In Arkavathy watershed, there are two stream gauging stations (namely Kanakpura and T. Bekuppe) maintained by Central Water Commission (CWC) and about 75 groundwater observations wells maintained by Central Ground Water Board (CGWB). In fact, the state agencies in Karnataka are monitoring stream flow of Cauvery river basin since 1934. Therefore, it is not clear which data paucity the authors are referring to.

## 3 Selected literature review

The authors argue that most of the research in India linking human drivers to hydrological responses consider administrative (local, region or national) boundaries as unit of analysis. They further claim that little attempt has been made to analyse





hydrological changes at the watershed, sub-basin and basin scale (page 596 of Penny et al., 2018). It implies that their study is perhaps the first such attempt to do so. Under this assumption, a large body of peer-reviewed research work already undertaken not only at the watershed scale (for instance by Garg et al., 2012; Syme et al., 2012), but also at the basin level (for instance by Gosain et al., 2010; Kumar, 2010) has been overlooked. Some of these studies are highly empirical in nature

and covered Sabarmati river basin (Kumar and Singh, 2001); Narmada river basin (Kumar et al., 2005; Kumar 2010), Ganga river basin (Anand et al., 2017), and, Brahmaputra and Kosi basins (Gosain et al., 2010). It would have been much more beneficial to the scientific community around the world if Penny et al. (2018) had analysed the existing studies on river basin management in India by including a critique on the approach, methodology and findings of those studies if they find them to be lacking in any way.

**4 Selecting appropriate unit of analysis**

From the discussions presented in the previous sections, it is clear that reliable official data on hydro-meteorology and groundwater for Arkavathy watershed is available for a long time period with various agencies and they are monitoring and managing data well. To assess the hydrological changes, Penny et al. (2018, page 598 and 600) had considered changes in water spread area in the tanks where inflows are not gauged and not the large reservoirs in the watershed of where inflows

are gauged, the reason being that reservoirs are actively managed for providing water for urban and agricultural uses. The whole purpose of using this approach was to identify changes in tank water extent which can be attributed to changes in tank inflows. Authors have used volumetric water balance equation for estimating the tank inflows (equation 1 on page 600 in Penny et al., 2018).

Instead, the large reservoirs located in the watershed and which are gauged could have been a much better choice to analyse

the hydrological changes due to changes in land use. A mass balance equation could have been used to estimate the actual inflows based on the available data on change in storage and outflows at different time intervals as data on rainfall at the reservoir site, releases from the dam, water losses and reservoir water level are available with the water resources department of the state. This data is also available in digitised form with the Karnataka State WDC, which is operated by Water Resources Department, Government of Karnataka for surface water and Groundwater Directorate, Karnataka for

groundwater (Table 1). Thus use of tanks, which are un-gauged, as a unit of analysis has actually adversely affected the confidence level of the model outputs and increased the uncertainty in the results as the tank water spread area may not be an accurate representation of the hydrological alterations happening in the watershed caused by land use changes and other factors, due to the reason that many complex factors (such as the nature of storage-area-elevation-curve of the tank) over and above the inflows affect the water spread area and these factors can change from location to location within the watershed.

**5 Assumptions and inferences**

Some assumptions and inferences in the paper by Penny et al. (2018) are somewhat hard to comprehend. A few of them are discussed herewith. First, a water balance equation is used to estimate the inflow into the ungauged tanks. For this, it is



assumed that the initial storage in the tank is zero (page 600 of Penny et al., 2018). However, this can only be true during hydrological years with low rainfall and also cannot be generalized for all the tanks.

Second, the authors have assumed outflows from the tank to be negligible (page 600-601 of Penny et al., 2018). This is based on observations of a few tanks and may be true for years with low rainfall. In years with normal and high rainfall,
tanks will have overflow which usually enters the downstream tank in cascade, a common occurrence in southern India. In Arkavathy watershed, there is very high inter-annual variability in rainfall. The rainfall can be as high as 1400 mm against an average rainfall of about 800mm (Fig. 2). Given the rainfall-runoff relationship for semi-arid areas (like Arkavathy watershed) wherein every unit increase in rainfall yield disproportionately higher increase in runoff (Kumar et al., 2006), the situation vis-à-vis tank inflows will be very different in wet years than that of dry years.

Third, the authors found that the variability in tank water extent due to precipitation across clusters spread throughout the Arkavathy watershed was similar (indicating no spatial variation in rainfall at the watershed scale) and for this they seem to have used rain gauging data for several locations (page 603 of Penny et al., 2018). However, our analysis on the average annual rainfall for 15 years (1998-2013) in Arkavathy watershed, using gridded rainfall data sets prepared by the India Meteorological Department and available on India-WRIS (at a high spatial resolution of 0.25º X 0.25º), shows spatial as well
as temporal variation in upper, middle and lower part of Arkavathy watershed (Fig. 2). The upper, middle and lower part represents the catchment area of three major tributaries of Arkavathy River. Between 1998 and 2013, the average annual rainfall varied from 377mm-1145mm (mean 794.5mm) in upper, 296mm-1422mm (mean 845mm) in middle and 376mm-1394mm (mean 896mm) in lower part of the watershed. The extent of inter-annual variability, as indicated by the coefficient of variation, is 29% for the upper, 40% for the middle, and 31% for the lower part of the watershed. At the micro-scale (e.g.
at tanks' catchment), the spatial variation in rainfall for catchments in different parts of the watershed will be even more and hence tanks in different parts of the watershed will show variation in their water extent.

Fourth, authors have mentioned that only limited data exists to describe historical declines in the groundwater table (page 605 of Penny et al., 2018). However, as was mentioned earlier, CGWB alone monitors about 50 observations wells in the area comprising Arkavathy watershed.

**6 Understanding groundwater behaviour**

The authors rely on the data collected from farmers to make an assessment on the groundwater levels in the basin (page 605 of Penny et al., 2018). Our contention is that farmers' data might be useful to understand the socio-economic aspects of local groundwater use but certainly not for understanding groundwater behaviour at the local or regional scale. Based on data collected from farmers, Penny et al. (2018) inferred that the groundwater level in Arkavathy watershed is declining. This is a
very sweeping inference as groundwater behaviour in hard rock formations (as in Arkavathy watershed) is a complex phenomenon. The water level in dug wells tapping weathered (unconfined) zone might not represent the regional ground water level if the rate of pumping is higher than the rate of recuperation of well, due to excessive drawdowns in water levels which is localized. Thus, for the purpose of understanding the groundwater balance due to rainfall and abstraction, the water





levels for measurement must essentially be the static water levels (as monitored by CGWB) and not any other dynamic water levels (as encountered in wells which are regularly pumped by farmers) due to 'unsteady-state' conditions that exist in the area surrounding the wells.

Contrary to the findings of Penny et al. (2018), the data of observation wells installed by CGWB that monitor groundwater
level in the basin indicate that the ground water fluctuation due to draft is positive in a major part of Cauvery middle sub-basin where Arkavathy lies. Analysis of long term trend in ground water levels using wells spread across Arkavathy watershed indicate that a higher proportion of observation wells recorded rise in water levels over the 20-year period (Fig. 3). The rising water level trend is likely to be in wells located downstream of urban centres like Bengaluru which receive its wastewater because of negative gradient with respect to surface water bodies. During non-monsoon months, most of the
inflows received by stream passing through Indian cities are wastewater. If the quantum of the flow leads to higher hydraulic head in the stream than in the groundwater system, it can result in flow of surface water to the aquifers and thus rise in groundwater levels. Jamwal et al. (2015) estimated that about 600 thousand cubic metres per day of wastewater flows from Bangalore city to Byramangala reservoir (downstream) in Arkavathy basin.

## 7 Conclusion

Surface water and groundwater interactions in river basins are quite complex and need good understanding of hydrology (rainfall, runoff) and geo-hydrology (groundwater level trends) and also the hydrological pressures and stresses (surface water diversion and groundwater draft) in the basin, to explain the cause-effect linkages scientifically (Kumar, 2010). Attempts using groundwater data collected from farmers and extrapolating it to watershed scale and using RS/GIS processed data without ground truthing will only yield misleading results. Since the watershed considered by Penny et al. (2018) is
gauged and long term data sets are available, their approach would have benefitted by comparing the results obtained from the distributed assessment (appropriate for regions characterised by data scarcity) with the integral assessment based on officially available long term data, to draw appropriate inferences about the hydrological dynamics in Arkavathy watershed.

**Competing interests.** The authors declare that they have no conflict of interest.

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





Table 1: Summary on Data Digitised by State Water Data Centre for various River Basins in Karnataka

| Observation Station | Type of Station | No of Observation Stations | Data Record Length | Remarks |
|---|---|---|---|---|
| Hydrological | Current meter gauging station | 32 | 1934 onwards for the old stations | Daily data on streamflow, rainfall, and reservoir storage, level, inflows and outflows can be obtained from the Hydrology Unit of Water Resources Development Organization, Government of Karnataka on submitting a written request explaining the purpose for which it is required. The requested data sets have to be obtained physically on payment of processing charges. |
| | Stage discharge station | 10 | | |
| | Silt and sedimentation observation station | 22 | | |
| Hydro-meteorological | Fully climatic station | 89 | 1970 onwards for the standard rain gauge stations | The website of Karnataka State Natural Disaster Monitoring Centre also maintains data (2011 onwards) on rainfall (https://www.ksndmc.org/Weather_info.aspx) and reservoir storage, level, inflows and outflows (https://www.ksndmc.org/Reservoir_Details.aspx). These data sets can be downloaded for free. |
| | Autographic rain gauge station | 206 | | |
| | Standard rain gauge station | 774 | | |
| Reservoir | Water level using radar level sensor | 16 | 2010 onwards for the stations set up under second phase of the Hydrology Project (2006-2014) | |
| Groundwater | Groundwater monitoring station (mainly Dug wells) | 1494 | Online data sets are available from 1996 onwards. Some Piezometers have been installed during | Groundwater level data can be obtained from the Groundwater Directorate (GD), Karnataka and Central Ground Water Board (CGWB), South Western Region, Bangalore, Karnataka. Data for observation wells under GD can be accessed on submitting a written request |
| | Piezometer | 442 | | |




the Hydrology Project (1996-2014)

explaining the purpose for which it is required. The requested data sets have to be obtained physically on payment of processing charges.

Data for observation wells under CGWB is available online and can be downloaded for free from the website of India-WRIS (http://www.india-wris.nrsc.gov.in/GWLevelApp.html?UType=R2VuZXJhbA==?UName=). A user account has to be created for downloading the data.

(Source: http://waterresources.kar.nic.in/wrdo.htm, http://www.india-wris.nrsc.gov.in/, and Central Ground Water Board, 2016)







Figure 1: Cauvery River Basin Map Showing Different Sub-basins and Spread of Hydrological Observation Sites maintained by CWC

5    (Source: Government of India, 2014)





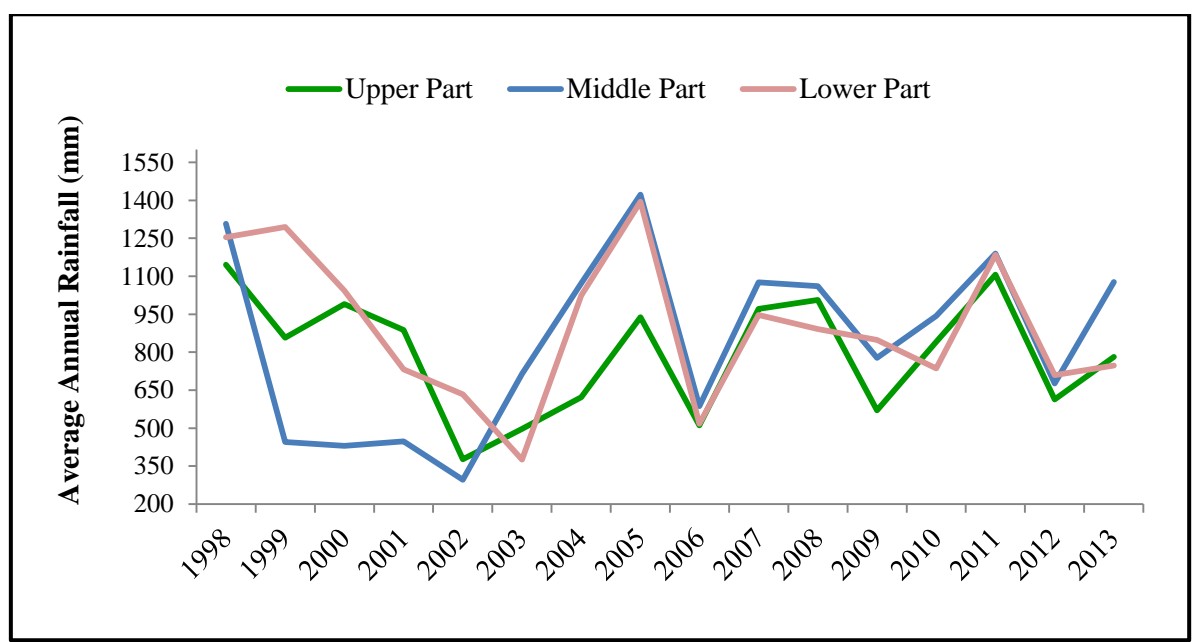

**Figure 2: Rainfall Variation in Arkavathy Watershed.**

(Source: Based on IMD gridded rainfall data sets)

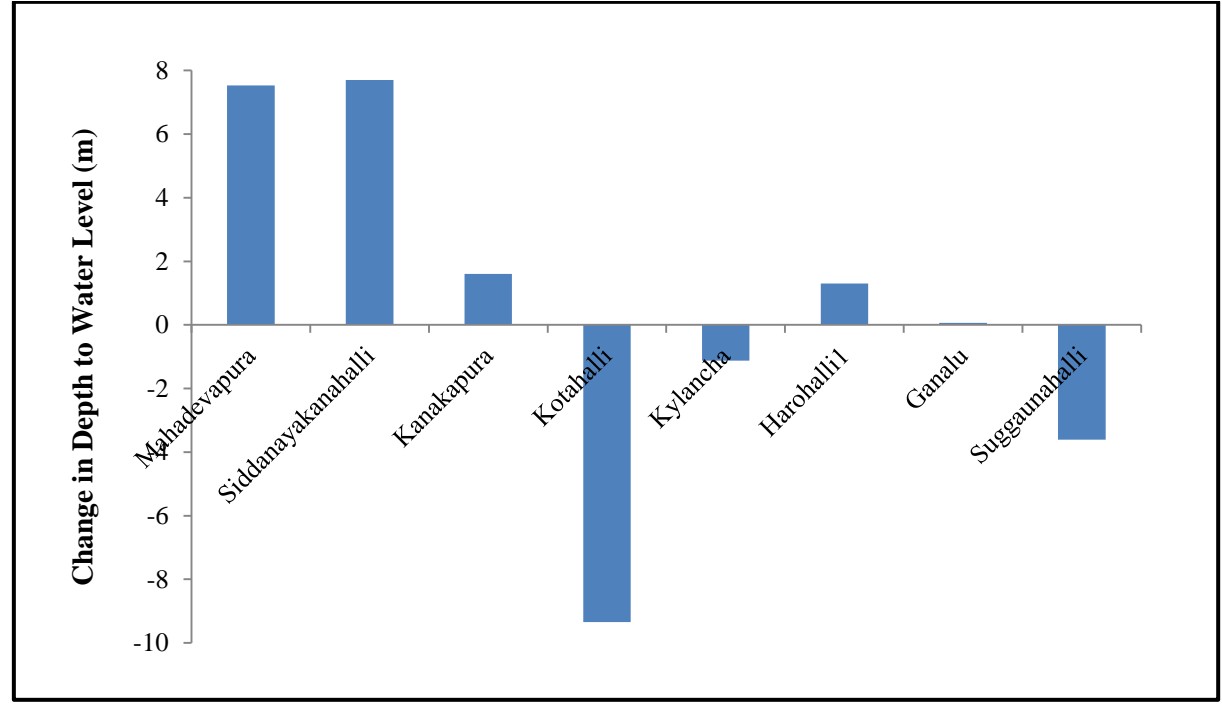

**Figure 3: Long Term Change in Groundwater Levels in Arkavathy Watershed: 1996-2015.**



(Source: Based on CGWB groundwater observation wells data sets)