# Peer review of "Comment on: "Spatial characterization of long-term hydrological change in the Arkavathy watershed adjacent to Bangalore, India" by Penny et al. (2018)"

_Hydrology and Earth System Sciences, 2018_

## Referee Comment (RC1) · Anonymous Referee #1 · 17 Oct 2018

Review of "Comment on: "Spatial characterization of long-term hydrological change in the Arkavathy watershed adjacent to Bangalore, India" by Penny et al. (2018)"

This manuscript presents some interesting comments on a "Spatial characterization of long-term hydrological change in the Arkavathy watershed adjacent to Bangalore, India" by Penny et al. (2018), previously published in HESS. The manuscript critiques the Penny paper on a number of points.

I am aware that a previous submission of this commentary has caused a large number of comments on the HESSD forum. However, I (on purpose) have not read the details of this conversation to remain as objective as possible in my current review.

The presented critiques on the Penny et al can be summarized as follows:

1) Various sources of ground data (streamflow, groundwater, precipitation, reservoir levels) exist, which are not acknowledged (or used) in the Penny et al. paper.

2) Past literature has discussed hydrological changes in this region, which, again, is not acknowledged in the Penny et al. paper.

3) Several assumptions in the Penny et al analyses are argued to be misleading.

These are interesting points that are may be relevant for HESS; especially highlighting the data availability of in this region helps to support future hydrological inferences of this region.

However, at present, the manuscript does not provide a structured scientific critique of the Penny et al paper that would lead to me recommending publishing this in HESS. Before I can recommend publication, the authors should address the following points:

[1] A critique of the Penny paper (in my opinion) requires tackling the main claims of their work. At present this critique does not clearly list the main claims of Penny et al., and therefore it also does not use the main findings of Penny et al. as the core of what needs to be "critiqued". In the context of the main current critiques, I expect that the authors show (or logically argue) how:

i) the data which is available leads to substantially different conclusions on the key findings of Penny, (ii) how past literature relates to the main findings of Penny et al, (iii) how a more realistic representation of reservoirs (i.e. section 4) affects the main findings of Penny et al, and (iv) how a better representation of the main assumptions and inferences of Penny et al that are listed in section 5, affect their main findings:

Properly addressing this point requires: (i) a better list of the main findings of Penny et al., (ii) a quantitative analysis (or very structured reasoning) that uses "better assumptions" and the available data to yield different conclusions than Penny et al.

[Figure]

Once such changes are made (if feasible), this critique could be changed into something that opens by an abstract that looks something like:

"Recent work by Penny et al. (2018) quantifies hydrological changes in the Arkavathy watershed, India, and finds that [LIST MAIN CLAIMS OF PENNY]. Here, we show how using local data and more realistic assumptions reveals that [LIST REVISED CLAIMS]."

[2] In many instances, the critique discusses the analysis of the Penny et al. paper without providing context for someone that is not familiar with all details of the Penny et al. work. For this critique to be more suitable for publication, I recommend that you always assume that the reader is not familiar with the details of this work, and you write the critique as a work that can be read as a stand-alone piece. In the list of detailed suggestions provided below, I have highlighted cases where sufficient background was lacking. However, I am unsure if this list is comprehensive, so please consider the entire text on this aspect in revising the paper.

[3] the overall quality of writing leads to many cases of inaccurate or unclear statements. IIn the detailed comments, I have highlighted where this is the case. However, I am unsure if this list is comprehensive, so please consider the entire text on this aspect in revising the paper.

Detailed comments

PAGE 1

L9-14: see suggestion for reframing the work according to the major comment provided above, and the suggestion that the abstract will be much more useful if you state it something like: ""Recent work by Penny et al. (2018) quantifies hydrological changes in the Arkavathy watershed, India, and finds that [LIST MAIN CLAIMS OF PENNY]. Here, we show how using local data and more realistic assumptions reveals that [LIST REVISED CLAIMS]."

L16: replace "more so" by "especially".

[Figure]

L16: The increase in water demand is independent of how arid a place is (i.e. the Moon is an arid place, but water DEMAND is very low, and not changing). Maybe you want to state something about water "stress" or a statement about "scarcity"? However, at present this argument is inaccurate. Please rephrase.

L17-18: "closure" is unclear wording in "are on the verge of closure or are already closed, with". (I understand what you mean, but why not simply replace it by "have". Also, do you have any references supporting this statement?

L20: Maybe "rivers and aquifers" is better since "aquifers" are parts of catchments"?

L20-24: this explanation seems redundant. In case you'd you like to keep it, please rephrase the text as you only cover part of the potential causes of hydrological changes.

L24: "In addition to" or "apart from"?

L25-26: "as most of the time it is the people in the latter that get affected by the hydrological changes occurring upstream." seems like a statement that can be supported by a reference?

PAGE 2

L1: "thus" is not warranted here. "Which suggests" or "which may lead to" seems more appropriate.

L5: "A few recent" or "Several"?

L5-7: Your wording suggests that this is an issue that is more wide-spread than the two citations that are listed here. Providing a more comprehensive reference list would be useful.

L7: "the official agencies" suggests you refer to specific agencies. Do you mean "official agencies" in a broader sense? If that is the case, remove "the".

L8: "are debated among the researchers." do you mean "discussed in person" or "discussed on published works". If the latter is the case, please cite some examples.

L8: Please cite the studies which refer to "In some cases"

L18-29: So if these data are available, show the reader how if they lead to the same or differences inferences than the main points of the Penny et al paper.

Entire section 3: state HOW previous literature relates to their work, rather than just stating they did mention past works in this regions

PAGE 3

L31: "somewhat hard to comprehend" is vague. Can you be more specific?

Section 4: "surface water extent" seems clearer than "water spread area".

PAGE 4

L3-9: Assuming zero outflow sounds naïve, but does it really matter given Penny et al state that when overflows occur "S is equal to its maximum Smax, so that variations in overflow cannot contribute to changes in observed S"? Thus, are these overflows not (implicitly) accounted for by having defined a maximum storage?

L10: I would change section heading because "unit" implies that the authors made some errors in the dimensions of the analysis (which is not the case).

L10-20: It is fine to show precipitation time-series, but it would be much more useful to actually calculate how these precipitation time-series affect reservoir behavior. Reservoirs buffer variations in rainfall and therefore do not necessarily show the same place-to-place variations as observed in the rainfall time series.

L22-24: Since these data exist, show us what these data look like and state if they are consistent, or contradicting, the findings of Penny et al.

L26-27: "Our contention is that farmers' data might be useful to understand the socio-economic aspects of local groundwater use but certainly not for understanding groundwater behaviour at the local or regional scale." It may be more fruitful to say "We explain how" rather than "Our contention is"...

L30: "very sweeping inference" why "sweeping" in this context?

PAGE 5

L4-6: "Contrary to the findings of Penny et al. (2018), the data of observation wells installed by CGWB that monitor groundwater 5 level in the basin indicate that the groundwater fluctuation due to draft is positive in a major part of Cauvery middle subbasin where Arkavathy lies." If this is the case please SHOW these data, rather than just state it.

L9: "because of the negative gradient with respect to surface water bodies" do you mean something like "because they are located at lower elevations than surrounding surface water bodies"?

L9-10: "inflows received by stream passing through Indian cities are wastewater." If this is the case, do you have any supporting this statement, or is this statement based on your local expert knowledge?

L10: what do you mean by "the quantum of the flow"?

L9-13: This is an interesting statement, but, at present, it seems to rely purely on speculation, rather than any data supporting that this is happening...

L15: "quite" seems redundant

L19: "will only yield misleading results" is a purely speculative (and very likely to be wrong) statement.

L15-22: Similar to the changes that are needed for the abstract. Rather than stating that Penny et al are wrong, SHOW how they are wrong

332, 2018.

---

## Referee Comment (RC2) · Anonymous Referee #2 · 8 Nov 2018

Review of Comment on: "Spatial characterization of long-term hydrological change in the Arkavathy watershed adjacent to Bangalore, India" by Penny et al. (2018) by Nitin Bassi, B.K. Harish Kumara, Meera Sahasranaman, Arijit Ganguly

General comments: The authors of this comment provide a critique which mainly centers on highlighting the "choice of inappropriate methodology (for a watershed which is actually gauged) and faulty assumptions by Penny et al. (2018) for analysing the watershed scale hydrological changes, generating misleading results and inferences". These are very strong words and I have read the comment as well as the original paper by Penny et al. and the supplementary material carefully to evaluate the arguments

brought to the table by this comment. I have to say that while I agree that it would have been good to compare the results of the distributed remote sensing based study with the observation well data that Bassi et al. refer to, there are also some arguments that seem to be based on misunderstandings. It would be important to take another close look at these arguments, check their validity and if possible show how the conlusions of Penny et al. are wrong before the comment is published. If in the end all that is left is the statement that Penny et al. should have made the effort of obtaining the observation well data I am also not sure if this really warrants the publication of this sort of comment. In this case a comment during the discussion phase of the manuscript by Penny et al. would have probably been more useful. In general I think the comment would profit from a slightly less confrontational tone, especially as some arguments seem to be based on misunderstandings. Actually showing how Penny et al. came to the wrong conclusions concerning the spatial patterns and their links to land use by doing a similar analysis based on the data provided by Bassi et al. would make this comment much stronger, but it is not clear if the coverage and spatial resolution as well as the length of the time series is sufficient to actually do this. The two stream gauges do not seem sufficient, but you could try to do this with the 75 observation wells.

Specific comments:

p.1 l.24: what do you mean by "aggregate level changes"? Please rephrase or explain.

p.1 l.25: please clarify what spatial patterns you are referring to here. Do you mean the spatial patterns of long-term changes in runoff? Or the aggregate level changes? As I do not understand what you mean by that it would be good to phrase this more clearly.

p.1.l.29: It is not uncommon to use methods for ungauged catchments on actually gauged catchments, either because the spatial and temporal resolution of the existing measurement data is insufficient for the purpose or because the data cannot be attained. In this case Penny et al. wanted to investigate the spatial pattern of runoff changes at a higher resolution than the official stream gauges provided.

p.2 l.8: "Often, existing resources with the official agencies to process and manage data properly are debated among the researchers" this sentence is unclear and needs to be rephrased.

p.2 l.17: what does it mean, a "fully operational WDC"? What do they do exactly? I tried to find more information on the internet, but could not find their webpage or a webpage describing their services. It is mentioned here https://www.karnataka.gov.in/karhp/Pages/Hydrology-Project-I.aspx, but this doesn't provide details on how to obtain data. It would be helpful if you could provide more information here.

p.2. l.24/25: I do not think it is very helpful to provide the numbers of observation sites of the Cauvery river basin here, please only focus on the study area of Penny et al., the Arkavathy catchment.

p.2. l.29: "Therefore, it is not clear which data paucity the authors are referring to." These are quite harsh words, given the fact that Penny et al. wanted to study the spatial patterns of hydrological changes in the 4250km$^2$ Arkavathy catchment. This is difficult to do if, as you state, only 2 stream gauges exist. It is admirable that there are streamflow time series starting from 1934 for the 81000km$^2$ Cauvery River, but this is also not helping much with the issue of spatial patterns in the Arkavathy catchment. I agree that it might have been helpful for the study of Penny et al. to compare their results with the 75 time series of groundwater observation wells. However, you say that this data is only available from 1996 onwards, while the Penny et al. study is focusing on changes for a longer time period between 1973 to 2010. In Table 1 you state "The requested data sets have to be obtained physically on payment of processing charges". What does this mean, "obtained physically"? Please clarify. Maybe you can also provide the information on how much the processing charges are? Please also provide the length of the time series for the two gauging stations in the Arkavathy Catchment.
p.2/3 l. 31-2: I do not find this statement in the text of Penny et al. They state that "There is little research that addresses the emergent effects and heterogeneity of human-driven hydrological change across the watershed scales at which management decisions must typically be made. The gap in scientific understanding at management relevant scales is strongly associated with a lack of data resolution at these scales, and..." which to me refers more to the spatial resolution that is necessary to capture the heterogeneity of the patterns. I would therefore suggest to rewrite or omit this sentence as it seems to be based on a misunderstanding.

p.3 l.3-6: do these studies look at the spatial patterns of hydrological response within the catchments at high resolution? Please add this information.

p.3 l.13: Penny et al. wanted to assess the spatial pattern of the hydrological changes, not simply the integral of hydrological changes. Please rephrase and reconsider your arguments under this aspect. I think this is where the main misunderstanding lies.

p.3 l.19: There are only 5 of these reservoirs, which is probably not enough to capture the pattern and variability the authors were after.

p.4 l.1.: This is discussed in Penny et al. p. 601

p.4. l.3: or happening at S=Smax, so variations in overflow do not produce variations in S. (as stated in Penny et al.). Please do not oversimplify when referencing the original study, this actually weakens your arguments.

p.4. l. 12: You state "Third, the authors found that the variability in tank water extent due to precipitation across clusters spread throughout the Arkavathy watershed was similar (indicating no spatial variation in rainfall at the watershed scale) and for this they seem to have used rain gauging data for several locations (page 603 of Penny et al., 2018)." Penny et al. used rain gauge data from 62 locations and looked at trends from 1971-2010. The spatial variability of rainfall is resolved through the use of the large number of raingauges. They show their results of the precipitation analysis in the

supplement and state that there is significant temporal variability but no temporal trend, so the same result that you confirm in Figure 2 and p.4 l. 12-19. If I have misunderstood your argument here, please rephrase and clarify.

p.5 l. 8: "The rising water level is likely to be in wells located downstream of urban centers…" I don't understand this statement. Which of the wells are located downstream of urban centers? This is something you know, not something you have to speculate on. Please clarify.

p.5 l. 10: The issue of wastewater increasing river water levels and leading to seepage into the aquifer: This seems to me to be an issue that is not really related to decreasing water storage due to irrigation with groundwater. And isn't this mainly affecting wells and aquifers directly adjacent to the rivers downstream of the urban centers? This sounds like a more local effect. What about groundwater elsewhere?

Table 1: please only focus on the Arkavathy Catchment and the data available here, as this is the point of the comment. Providing measurement locations for the entire state of Karnataka is confusing and not really helping the discussion.

Figure 1: The figure is too small and not readable. I suggest focusing on the Arkavathy Catchment as this is where the study of Penny et al. took place. Also include the location of the 75 observation wells, possibly also indicating the length of the respective time series.

Figure 3: Is this based on simply comparing the annual mean water levels in 1996 and 2015? It might be more interesting to supply the actual time series. Please also provide the locations of these wells in the map of the Arkavathy Catchment (revised Figure 1). It would also be more useful to actually show the entire data set of groundwater time series of all 75 observations wells. That would be much more convincing than showing just a selected few.

332, 2018.

---

## Author Comment (AC1) · 20 Dec 2018

Thank you so much to the referee for the constructive comments. All the comments have been addressed and are detailed below.

This manuscript presents some interesting comments on a "Spatial characterization of long-term hydrological change in the Arkavathy watershed adjacent to Bangalore, India" by Penny et al. (2018), previously published in HESS. The manuscript critiques the Penny paper on a number of points.
I am aware that a previous submission of this commentary has caused a large number of comments on the HESSD forum. However, I (on purpose) have not read the details of this conversation to remain as objective as possible in my current review.

**Response:** Thank you so much for your observations and pragmatic approach.

The presented critiques on the Penny et al can be summarized as follows:
1) Various sources of ground data (streamflow, groundwater, precipitation, reservoir levels) exist, which are not acknowledged (or used) in the Penny et al. paper.
2) Past literature has discussed hydrological changes in this region, which, again, is not acknowledged in the Penny et al. paper.
3) Several assumptions in the Penny et al analyses are argued to be misleading.

These are interesting points that are may be relevant for HESS; especially highlighting the data availability of in this region helps to support future hydrological inferences of this region.

**Response:** Thank you for the feedback.

However, at present, the manuscript does not provide a structured scientific critique of the Penny et al paper that would lead to me recommending publishing this in HESS. Before I can recommend publication, the authors should address the following points:

**Response:** Sure, we have provided our responses to all the queries raised.

[1] A critique of the Penny paper (in my opinion) requires tackling the main claims of their work. At present this critique does not clearly list the main claims of Penny et al., and therefore it also does not use the main findings of Penny et al. as the core of what needs to be "critiqued". In the context of the main current critiques, I expect that the authors show (or logically argue) how:
i) the data which is available leads to substantially different conclusions on the key findings of Penny,
(ii) how past literature relates to the main findings of Penny et al,
(iii) how a more realistic representation of reservoirs (i.e. section 4) affects the main findings of Penny et al, and
(iv) how a better representation of the main assumptions and inferences of Penny et al that are listed in section 5, affect their main findings:
Properly addressing this point requires: (i) a better list of the main findings of Penny et al., (ii) a quantitative analysis (or very structured reasoning) that uses "better assumptions" and the available data to yield different conclusions than Penny et al.
Once such changes are made (if feasible), this critique could be changed into something that opens by an abstract that looks something like:
"Recent work by Penny et al. (2018) quantifies hydrological changes in the Arkavathy watershed, India, and finds that [LIST MAIN CLAIMS OF PENNY]. Here, we show how using local data and more realistic assumptions reveals that [LIST REVISED CLAIMS]."

**Response:** Thanks for sharing this feedback. In response to the above comments, we would like to state the following:

1] Our aim is to highlight the limitations of the methodological approach followed by Penny et al which is more suitable for the un-gauged catchment. In this reference only, we have explained how some of the assumptions and inferences (section 6) considered by Penny et al, for a watershed which is gauged, are factually incorrect. To this, we have provided analysis using secondary data (rainfall and groundwater levels) and literature.

2] The onus of relating with the existing literature in terms of approach, methodology and findings of those studies was with Penny et al. which they did not attempt. We have highlighted this in section 3 of the original manuscript.

3] Even If Penny et al. wanted to test their methodology for the academic purpose, the results should have been compared with those obtained using the available official data sets. The point we have highlighted in the conclusion section too.

4] In the abstract too, we have clearly highlighted the problems with their methodological approach. In fact, we finalised the abstract only after incorporating comments from the Editor.

[2] In many instances, the critique discusses the analysis of the Penny et al. paper without providing context for someone that is not familiar with all details of the Penny et al. work. For this critique to be more suitable for publication, I recommend that you always assume that the reader is not familiar with the details of this work, and you write the critique as a work that can be read as a stand-alone piece. In the list of detailed suggestions provided below, I have highlighted cases where sufficient background was lacking. However, I am unsure if this list is comprehensive, so please consider the entire text on this aspect in revising the paper.

**Response:** Thanks for providing a detailed list of suggestions. We have attended to all such suggestions, and will also do one more round of check before submitting the final revised manuscript.

[3] the overall quality of writing leads to many cases of inaccurate or unclear statements. In the detailed comments, I have highlighted where this is the case. However, I am unsure if this list is comprehensive, so please consider the entire text on this aspect in revising the paper.

**Response:** Thanks, we have addressed all such queries, and will also do one more round of check before submitting the final revised manuscript.

**Detailed comments**
PAGE 1
L9-14: see suggestion for reframing the work according to the major comment provided above, and the suggestion that the abstract will be much more useful if you state it something like: ""Recent work by Penny et al. (2018) quantifies hydrological changes in the Arkavathy watershed, India, and finds that [LIST MAIN CLAIMS OF PENNY]. Here, we show how using local data and more realistic assumptions reveal that [LIST REVISED CLAIMS]."

**Response:** We have already explained in one of the previous responses that our aim is to highlight the flaws with the methodological approach followed by Penny et al. and the abstract clearly states it.

L16: replace "more so" by "especially".

**Response:** Thanks, it will be replaced in the revised manuscript.

L16: The increase in water demand is independent of how arid a place is (i.e. the Moon is an arid place, but water DEMAND is very low, and not changing). Maybe you want to state something about

water "stress" or a statement about "scarcity"? However, at present this argument is inaccurate. Please rephrase.

**Response:** We have argued that globally there has been increase in water demand for countries in arid and semi-arid regions. We are not saying that aridity results in increase in water demand but it surely induces water stress. For instance, in India, most of the agricultural prosperous regions (North-western and western India) are semi-arid to arid with increasing water demand for agriculture being now met or planned to be met through inter-basin water transfers. Indira-Gandhi Nahar Project and Sardar Sarovar Narmada Project are some examples. We will make the argument clearer in in the revised manuscript.

L17-18: "closure" is unclear wording in "are on the verge of closure or are already closed, with". (I understand what you mean, but why not simply replace it by "have". Also, do you have any references supporting this statement?

**Response:** Agreed, we will revise the statement in the revised manuscript. Sure, we will also provide the reference for the same (IWMI published some studies).

L20: Maybe "rivers and aquifers" is better since "aquifers" are parts of catchments"?

**Response:** Here we have used term catchment as it may have structures to harvest surface runoff. Hence, rivers cannot be considered as the only source of surface water.

L20-24: this explanation seems redundant. In case you'd you like to keep it, please rephrase the text as you only cover part of the potential causes of hydrological changes.

**Response:** We would like to keep it as it informs readers on the potential reasons for long-term changes in runoff and groundwater availability. In our opinion, this is very much related to the theme of article being commented upon.

L24: "In addition to" or "apart from"?

**Response:** As suggested, we will make a change in the revised manuscript.

L25-26: "as most of the time it is the people in the latter that get affected by the hydrological changes occurring upstream." seems like a statement that can be supported by a reference?

**Response:** Yes, we will provide reference/s in support of this statement in the revised manuscript. Thanks for pointing this out.

PAGE 2

L1: "thus" is not warranted here. "Which suggests" or "which may lead to" seems more appropriate.

**Response:** 'This suggests' seems to be better, in any case we will make the correction in the revised manuscript.

L5: "A few recent" or "Several"?

**Response:** India has made tremendous progress with its water resources data management system, India-WRIS being a perfect example of it. Researchers are regularly using data sets and information (on Hydrology, groundwater, and meteorology) from India-WRIS. Thus, we would like to keep this sentence as it is.

L5-7: Your wording suggests that this is an issue that is more wide-spread than the two citations that are listed here. Providing a more comprehensive reference list would be useful.

**Response:** In our experience, in most of the cases such perceptions are from the researchers based in institutions outside India. It is usual that they may not have access to right sources of data and information. Nevertheless, we will explore whether we can find one or two more such studies.

L7: "the official agencies" suggests you refer to specific agencies. Do you mean "official agencies" in a broader sense? If that is the case, remove "the".

**Response:** Fine, we will remove 'the' in the revised manuscript.

L8: "are debated among the researchers." do you mean "discussed in person" or "discussed on published works". If the latter is the case, please cite some examples.

**Response:** Yes, we mean discussed in-person.

L8: Please cite the studies which refer to "In some cases"

**Response:** Thanks, we will add references in the revised manuscript.

L18-29: So if these data are available, show the reader how if they lead to the same or differences inferences than the main points of the Penny et al paper.

**Response:** We would like to clarify that our objective of the commentary is to highlight that when official validated data sets are available, why RS and GIS is being adapted as a methodological approach to assess hydrological changes? Because of the latter approach, Penny et al. has made some assumptions which are incorrect. This assessment is based on our experience of working in river basins in India and also on various other studies which have been cited in section 3. We also specify in the conclusion section that 'their approach would have benefitted by comparing the results obtained from the distributed assessment (appropriate for un-gauged catchments) with the integral assessment based on officially available long term data'.

Entire section 3: state HOW previous literature relates to their work, rather than just stating they did mention past works in this regions

**Response:** This is what we have suggested that they should have 'analysed the existing studies on river basin management in India by including a critique on the approach, methodology and findings of those studies if they find them to be lacking in any way' (L 7-9, section 3, page 3). The real issue is Penny et al. did not even attempt to relate to the existing studies.

PAGE 3
L31: "somewhat hard to comprehend" is vague. Can you be more specific?

**Response:** There are certain changes which have been made in consultation with the editor, this is one of them. We have no problem if it needs to be further revised.

Section 4: "surface water extent" seems clearer than "water spread area".

**Response:** In our opinion, when it comes to highlighting changes, water spread area appears more scientific. Extent denotes a range.

PAGE 4
L3-9: Assuming zero outflow sounds naïve, but does it really matter given Penny et al state that when overflows occur "S is equal to its maximum Smax, so that variations in overflow cannot contribute to changes in observed S"? Thus, are these overflows not (implicitly) accounted for by having defined a maximum storage?

**Response:** We have checked again on the 'outflow' assumption in Penny et al. They state 'variations in Qout can be neglected, for two reasons: first, because watershed managers report that tanks rarely overflow, so Qout (overflows leaving the tank) can reasonably be approximated as=0, and, second, because any overflow that does occur implies that S is equal to its maximum Smax, so that variations in overflow cannot contribute to changes in observed S'. There are two different reasons. We would like to clarify that our argument is for the first part of this statement that Qout can reasonably be approximated as=0. This assumption will certainly affect the results as Qout is one of the variables used for eq. 1 in Penny et al. Hence, we would like to retain this argument as it is.

L10: I would change section heading because "unit" implies that the authors made some errors in the dimensions of the analysis (which is not the case).

**Response:** We assume this is a suggestion for Section Title 4 on page 3. We will replace 'unit' with 'assessment unit' in the revised manuscript.

L10-20: It is fine to show precipitation time-series, but it would be much more useful to actually calculate how these precipitation time-series affect reservoir behavior. Reservoirs buffer variations in rainfall and therefore do not necessarily show the same place-to-place variations as observed in the rainfall time series.

**Response:** Thanks for sharing this observation. However, Penny et al. have used tanks as their assessment units which actually are much smaller in scale then reservoirs and thus have limited capacity to buffer variations in rainfall. We have highlighted this, please refer to lines 18-21 on page 4 of the original manuscript.

L22-24: Since these data exist, show us what these data look like and state if they are consistent, or contradicting, the findings of Penny et al.

**Response:** We have analysed the groundwater data for the Arkavathy watershed which has been presented in detail in section 6. Please refer to lines 4-13 on page 5 and Figure 3 on page 10 of the original manuscript.

L26-27: "Our contention is that farmers' data might be useful to understand the socioeconomic aspects of local groundwater use but certainly not for understanding ground- water behaviour at the local or regional scale." It may be more fruitful to say "We explain how" rather than "Our contention is":

**Response:** It is a reality based on our experience of working in hard rock regions of India, hence the word contention is used. We will replace 'Our contention' with 'Our experience' in the revised manuscript.

L30: "very sweeping inference" why "sweeping" in this context?

**Response:** Sweeping in two context: 1] As we explained in previous lines (27-28, page 4), recall or perception based data cannot be used for understanding groundwater behaviour at the local or regional scale; and, 2] As we have explained in lines 30-33 on page 4 and lines 1-3 on page 5, groundwater behaviour in hard rock formations (as in Arkavathy watershed) is a complex phenomenon, thus measurements must essentially be the static water levels (as monitored by CGWB, an official agency) and not any other dynamic water levels (as encountered in wells which are regularly pumped by farmers) due to 'unsteady-state' conditions that exist in the area surrounding the wells.

PAGE 5

L4-6: "Contrary to the findings of Penny et al. (2018), the data of observation wells installed by CGWB that monitor groundwater level in the basin indicate that the groundwater fluctuation due to draft is positive in a major part of Cauvery middle sub-basin where Arkavathy lies." If this is the case please SHOW these data, rather than just state it.

**Response:** We have presented the data set in Figure 3 on page 10 of the original manuscript.

L9: "because of the negative gradient with respect to surface water bodies" do you mean something like "because they are located at lower elevations than surrounding surface water bodies"?

**Response:** By the negative gradient we mean that the water level in wells is at the lower elevation with respect to the water level in the surface water bodies. We have further explained this phenomenon in lines 9-12 on page 5 of the original manuscript.

L9-10: "inflows received by stream passing through Indian cities are wastewater." If this is the case, do you have any supporting this statement, or is this statement based on your local expert knowledge?

**Response:** This is based on expert knowledge and as well as studies pointing to closed nature of river basins in Peninsular India (where Arkavathy watershed lies). We have already agreed to provide reference with respect to the closed nature of these river basins.

L10: what do you mean by "the quantum of the flow"?

**Response:** It refers to the amount of water, we will make it explicit in the revised version of the manuscript.

L9-13: This is an interesting statement, but, at present, it seems to rely purely on speculation, rather than any data supporting that this is happening:

**Response:** As indicated in one of the previous responses, this is based on our local expert knowledge on the hydrology and geo-hydrology of the Peninsular India which is mostly underlain by hard rocks. In this region, wells go dry during summers (seasonal groundwater scarcity) but surface water bodies continue to receive wastewater from the city. We have provided data on amount of wastewater

which flows from Bangalore city to one of the reservoirs in Arkavathy watershed (line 12-13, page 5). This leads to higher hydraulic gradient in the stream then in the groundwater (as wells are dry). We will further strengthen the paragraph by including discussion on the seasonal groundwater scarcity aspects and providing some references for the same.

L15: "quite" seems redundant

**Response:** We will remove it in the revised manuscript.

L19: "will only yield misleading results" is a purely speculative (and very likely to be wrong) statement.

**Response:** Misleading in terms of approach which we have discussed all along our commentary. We will make it clear in the revised manuscript.

L15-22: Similar to the changes that are needed for the abstract. Rather than stating that Penny et al are wrong, SHOW how they are wrong

**Response:** We will again emphasize, our commentary is on the approach followed by Penny et al., which we have discussed in detail. Even the conclusion (lines 19-23 on page 5) ends with the same note.

---

## Author Comment (AC2) · 20 Dec 2018

Thank you so much to the referee for the constructive comments. All the comments have been addressed and are detailed below.

General comments: The authors of this comment provide a critique which mainly centers on highlighting the "choice of inappropriate methodology (for a watershed which is actually gauged) and faulty assumptions by Penny et al. (2018) for analysing the watershed scale hydrological changes, generating misleading results and inferences". These are very strong words and I have read the comment as well as the original paper by Penny et al. and the supplementary material carefully to evaluate the arguments brought to the table by this comment. I have to say that while I agree that it would have been good to compare the results of the distributed remote sensing based study with the observation well data that Bassi et al. refer to, there are also some arguments that seem to be based on misunderstandings. It would be important to take another close look at these arguments, check their validity and if possible show how the conclusions of Penny et al. are wrong before the comment is published. If in the end all that is left is the statement that Penny et al. should have made the effort of obtaining the observation well data I am also not sure if this really warrants the publication of this sort of comment. In this case a comment during the discussion phase of the manuscript by Penny et al. would have probably been more useful. In general I think the comment would profit from a slightly less confrontational tone, especially as some arguments seem to be based on misunderstandings. Actually showing how Penny et al. came to the wrong conclusions concerning the spatial patterns and their links to land use by doing a similar analysis based on the data provided by Bassi et al. would make this comment much stronger, but it is not clear if the coverage and spatial resolution as well as the length of the time series is sufficient to actually do this. The two stream gauges do not seem sufficient, but you could try to do this with the 75 observation wells.

**Response:** Thanks for the suggestions. On the data sets, we mentioned on the data availability with the Central agencies, State data centre has data for many more observations stations/sites (table 1) and as mentioned one has to visit them in-person to understand the status in Cauvery basin, in this case for Arkavathy watershed. Whereas Penny et al have used a methodology on the notion that data sets are not available and hence in all likelihood did not made any effort to consult the state data management agencies. Through this commentary, our aim is also to highlight the limitation of using a methodology (for a gauged catchment) which is more suitable for an ungauged-catchment. Onus, of comparing the results with the integral assessment based on officially available long term data was with Penny et al. which we have highlighted.
On the tone and language used, the commentary has been edited several times and has been put forth for discussion only after incorporating suggested changes by the editor.

Specific comments:

p.1 l.24: what do you mean by "aggregate level changes"? Please rephrase or explain.

**Response:** In this case, it refers to the hydrological changes at the watershed scale. As suggested, we will elaborate this in the revised manuscript.

p.1 l.25: please clarify what spatial patterns you are referring to here. Do you mean the spatial patterns of long-term changes in runoff? Or the aggregate level changes? As I do not understand what you mean by that it would be good to phrase this more clearly.

**Response:** Actually, it is spatial patterns of long term changes in runoff and groundwater availability. We would make the statement clear in the revised manuscript

p.1.l.29: It is not uncommon to use methods for ungauged catchments on actually gauged catchments, either because the spatial and temporal resolution of the existing measurement data is insufficient for the purpose or because the data cannot be attained. In this case Penny et al. wanted to investigate the spatial pattern of runoff changes at a higher resolution than the official stream gauges provided.

**Response:** Sure, for this reason we have concluded the commentary by arguing that Penny et al should have compared the 'results obtained from the distributed assessment (appropriate for regions characterised by data scarcity) with the integral assessment based on officially available long term data, to draw appropriate inferences about the hydrological dynamics in Arkavathy watershed'. At least, an attempt should have been made to know the extent of official data availability.

p.2 l.8: "Often, existing resources with the official agencies to process and manage data properly are debated among the researchers" this sentence is unclear and needs to be rephrased.

**Response:** We simply mean it is 'often discussed among researchers in-person'. We will clarify this in the revised manuscript.

p.2 l.17: what does it mean, a "fully operational WDC"? What do they do exactly? I tried to find more information on the internet, but could not find their webpage or a webpage describing their services. It is mentioned here https://www.karnataka.gov.in/karhp/Pages/Hydrology-Project-I.aspx, but this doesn't provide details on how to obtain data. It would be helpful if you could provide more information here.

**Response:** 'WDC' refers to Water Data Centre (already mentioned in the previous line of the original manuscript) and 'fully operational' means that they are collecting, validating and processing (for further use) all the state hydrological data sets. For accessing the required data sets, one has to approach them formally. This has been already mentioned in detail in remarks column of Table 1 on page 7 of the original manuscript.

p.2. l.24/25: I do not think it is very helpful to provide the numbers of observation sites of the Cauvery river basin here, please only focus on the study area of Penny et al., the Arkavathy catchment.

**Response:** Since the whole debate is about the availability or non-availability of data, it is important to provide a larger picture in terms of data availability at the basin scale which can also be of use to other researchers who are interested in working on issues in Cauvery river basin.

p.2. l.29: "Therefore, it is not clear which data paucity the authors are referring to." These are quite harsh words, given the fact that Penny et al. wanted to study the spatial patterns of hydrological changes in the 4250km2 Arkavathy catchment. This is difficult to do if, as you state, only 2 stream gauges exist. It is admirable that there are streamflow time series starting from 1934 for the 81000km2 Cauvery River, but this is also not helping much with the issue of spatial patterns in the Arkavathy catchment. I agree that it might have been helpful for the study of Penny et al. to compare their results with the 75 time series of groundwater observation wells. However, you say that this data is only available from 1996 onwards, while the Penny et al. study is focusing on changes for a longer time period between 1973 to 2010. In Table 1 you state "The requested data sets have to be obtained physically on payment of processing charges". What does this mean, "obtained physically"? Please clarify. Maybe you can also provide the information on how much the processing charges are? Please also provide the length of the time series for the two gauging stations in the Arkavathy Catchment.

**Response:** There are reservoirs in the Arkavathy watershed as well. Nevertheless, we have already explained in detail in the original manuscript as well in the response to one of the previous comments regarding the data availability in the Arkavathy watershed and also on the fact that onus of comparing results (remote sensing based distributed assessment with integral assessment using official data sets) was with Penny et al.

Obtain physically means that one has to go there in-person and make a request formally. Payment charges will depend on the data sets requested which only the officials can tell.

As suggested, we will mention the time series of the data sets (available from 1979 onwards) for the two gauging stations in the revised manuscript.

p.2/3 l. 31-2: I do not find this statement in the text of Penny et al. They state that "There is little research that addresses the emergent effects and heterogeneity of human-driven hydrological change across the watershed scales at which management decisions must typically be made. The gap in scientific understanding at management relevant scales is strongly associated with a lack of data resolution at these scales, and..." which to me refers more to the spatial resolution that is necessary to capture the heterogeneity of the patterns. I would therefore suggest to rewrite or omit this sentence as it seems to be based on a misunderstanding.

**Response:** Thanks for the observation. As per our understanding, Penny et al. has argued that most of the research linking human drivers to hydrological responses considers administrative boundaries (Local, Regional, and National) rather than hydrological boundaries (watershed, sub-basin, and basin) as a unit of analysis. And that their study has considered the relevant hydrological scale. They are right that hydrological scales need to be considered but they should have acknowledged the contributions made by others. Thus the scale they are mentioning is for the hydrological unit (watershed, sub-basin or basin). This is what we have highlighted in lines 31-2 on page no. 2/3 in the original manuscript. Also, the argument has already been revised as per the suggestion of the editor.

p.3 l.3-6: do these studies look at the spatial patterns of hydrological response within the catchments at high resolution? Please add this information.

**Response:** As clarified above, the scale which Penny et al. is mentioning is for the hydrological unit (watershed, sub-basin, and basin). Accordingly, we have provided references in the original manuscript mentioning studies which have used empirical data to analyse the hydrological changes at the watershed and basin scale.

p.3 l.13: Penny et al. wanted to assess the spatial pattern of the hydrological changes, not simply the integral of hydrological changes. Please rephrase and reconsider your arguments under this aspect. I think this is where the main misunderstanding lies.

**Response:** As per our opinion, we do not have any misunderstanding. Perhaps, we should clarify again that our aim is to highlight the limitations of using the RS/GIS based methodology for the watershed which is gauged. Even in the conclusion, we have clearly mentioned that Penny et al. should have at least compared the results obtained through their distributed assessment by undertaking integral assessment using official data sets.

p.3 l.19: There are only 5 of these reservoirs, which is probably not enough to capture the pattern and variability the authors were after.

**Response:** That is why we mention that at least they should have compared the results. Penny et al has estimated changes in the surface areas of distributed tanks (using RS/GIS) and regarded them as

proxy for the balance of surface flows. In our opinion, such assessment are more uncertain than based on operational data (as mentioned in the original manuscript), depending on the precision of the satellite observations and on the degree to which the surface to volume ratio of the tanks is known. Such approach is useful for the ungauged areas. However, Arkavathy watershed is gauged. Therefore, we have mentioned in the original manuscript and state it again that at least they should have compared the results.

p.4 l.1.: This is discussed in Penny et al. p. 601

**Response:** We have checked again, Penny et al. have considered initial storage in all the tanks (which are spatially distributed) to be zero in spite of high inter-annual and spatial variability in rainfall (please refer lines 6-21 one page 4 of the original manuscript).

p.4. l.3: or happening at S=Smax, so variations in overflow do not produce variations in S. (as stated in Penny et al.). Please do not oversimplify when referencing the original study, this actually weakens your arguments.

**Response:** Thanks, we will revise the statement in the revised manuscript.

p.4. l. 12: You state "Third, the authors found that the variability in tank water extent due to precipitation across clusters spread throughout the Arkavathy watershed was similar (indicating no spatial variation in rainfall at the watershed scale) and for this they seem to have used rain gauging data for several locations (page 603 of Penny et al., 2018)." Penny et al. used rain gauge data from 62 locations and looked at trends from 1971-2010. The spatial variability of rainfall is resolved through the use of the large number of raingauges. They show their results of the precipitation analysis in the supplement and state that there is significant temporal variability but no temporal trend, so the same result that you confirm in Figure 2 and p.4 l. 12-19. If I have misunderstood your argument here, please rephrase and clarify.

**Response:** Our main argument is on the spatial variability of rainfall across the tank clusters which Penny et al has not considered. Apart from inter-annual variations, rainfall analysis presented in Figure 2 and discussed in the manuscript (lines 10-21, page 4 of the original manuscript) clearly show spatial variation in the rainfall in the upper, middle and lower parts of the watershed. Please further note that since the gridded rainfall data sets having high spatial resolution (0.25 X 0.25) is used, any variation has relevance even at the reservoir/tank scale (mainly between tanks located in different parts of the watershed).

p.5 l. 8: "The rising water level is likely to be in wells located downstream of urban centers: : :" I don't understand this statement. Which of the wells are located downstream of urban centers? This is something you know, not something you have to speculate on. Please clarify.

**Response:** This is to explain the groundwater behaviour using our local expert knowledge. Any future research study may explore on the surface-groundwater interactions in this watershed.

p.5 l. 10: The issue of wastewater increasing river water levels and leading to seepage into the aquifer: This seems to me to be an issue that is not really related to decreasing water storage due to irrigation with groundwater. And isn't this mainly affecting wells and aquifers directly adjacent to the rivers downstream of the urban centers? This sounds like a more local effect. What about groundwater elsewhere?

**Response:** Peninsular India (where Arkavathy lies) is mostly underlain by hard rocks. In this region, wells go dry during summers (seasonal groundwater scarcity) but surface water bodies continue to receive wastewater from the city. We have provided data on amount of wastewater which flows from Bangalore city to one of the reservoirs in Arkavathy watershed (line 12-13, page 5). This leads to higher hydraulic gradient in the stream then in the groundwater (as wells are dry). We will further strengthen the paragraph by including discussion on the seasonal groundwater scarcity aspects and providing some references for the same.

Table 1: please only focus on the Arkavathy Catchment and the data available here, as this is the point of the comment. Providing measurement locations for the entire state of Karnataka is confusing and not really helping the discussion.

**Response:** We have provided the details of the data availability in the Arkavathy watershed in lines 26-29 on page 2 of the original manuscript. The idea of presenting this table (which is also suggested by the Editor) is to highlight the data availability with the state water data centre. One has to visit them to understand how many of these locations are within the Cauvery basin or more specifically in the Arkavathy watershed. We have mentioned this in the remarks column of the Table 1.

Figure 1: The figure is too small and not readable. I suggest focusing on the Arkavathy Catchment as this is where the study of Penny et al. took place. Also include the location of the 75 observation wells, possibly also indicating the length of the respective time series.

**Response:** This map is prepared by the Central Water Commission and hence we are unable to provide more details. But surely, it informs readers and other interested researchers on the hydrological observations sites in the basin. For similar reasons, we are unable to provide the location of wells in the watershed. Nevertheless, for groundwater, time series is mentioned in the caption of Figure 3, and for the stream gauges, we have already agreed to provide the time series.

Figure 3: Is this based on simply comparing the annual mean water levels in 1996 and 2015? It might be more interesting to supply the actual time series. Please also provide the locations of these wells in the map of the Arkavathy Catchment (revised Figure 1). It would also be more useful to actually show the entire data set of groundwater time series of all 75 observations wells. That would be much more convincing than showing just a selected few.

**Response:** Figure 3 is based on comparing the observed (by CGWB) pre monsoon water level (to account for draft in a given year) data of each well. It is not the average. We have used only those wells for which data was available with us and these wells have been spread across Arkavathy watershed. Thus, spatial variation is captured, we can provide the lat-long of the wells in the revised manuscript. For data on all the 75 observation wells, one has to approach CGWB or Karnataka State Water Data Centre which we would not like to do for the commentary.